# Long-Term Lifestyle Habits and Quality of Life after Roux-in-Y Gastric Bypass in Brazilian Public versus Private Healthcare Systems: Beyond Weight Loss

**DOI:** 10.3390/ijerph20156494

**Published:** 2023-08-01

**Authors:** Mariana S. Melendez-Araújo, Ariene Silva do Carmo, Flávio Teixeira Vieira, Fernando Lamarca, Eduardo Yoshio Nakano, Ricardo M. Lima, Eliane Said Dutra, Kênia Mara Baiocchi de Carvalho

**Affiliations:** 1Graduate Program of Human Nutrition, University of Brasília, Brasília 70910-900, Brazil; nutmelendez@gmail.com (M.S.M.-A.); flaviovieira.nutri@gmail.com (F.T.V.); fernando.pardo@uerj.br (F.L.); ricardomoreno@unb.br (R.M.L.); elidutra@unb.br (E.S.D.); 2Ministry of Health, Rio de Janeiro 70068-900, Brazil; ariene.carmo@saude.gov.br; 3Department of Applied Nutrition, Rio de Janeiro State University (UERJ), Rio de Janeiro 23900-000, Brazil; 4Department of Statistics, University of Brasilia, Brasília 70910-900, Brazil; nakano@unb.br; 5Graduate Program in Physical Education, University of Brasilia, Brasília 70910-900, Brazil

**Keywords:** treatment outcome, public health, bariatric surgery, long-term care, quality of life, lifestyle habits

## Abstract

Lifestyle and health-related quality of life (HRQoL) are good markers of surgical obesity treatment. This study aimed to investigate the lifestyle and HRQoL of patients at least five years after Roux-en-Y gastric bypass in public (SUS) and private (PVT) Brazilian healthcare systems. In this cross-sectional study, weight loss (WL), % of excess WL (%EWL), diet quality, physical activity, alcohol consumption, and HRQoL were evaluated. Analysis of covariance, binary and multinomial logistic regression, adjusted for confounders, were performed. The SUS group had more vulnerable socioeconomic statuses than the PVT group. Total %WL and % EWL were 24.64 ± 0.99% and 60.46 ± 2.41%, respectively, without difference between groups. In the Pain/Discomfort and Anxiety/Depression domains of HRQoL, more than 50% reported moderate problems without differences between groups. Processed food ingestion was higher in the PVT (132.10 ± 60.15 g/1000 kcal) than in the SUS (103.43 ± 41.72 g/1000 kcal), however, without statistical significance (*p* = 0.093). The PVT group showed lower physical activity (OR: 0.23; 95%CI: 0.87–0.63; *p* = 0.004) and a higher risk of alcohol-related problems (OR: 3.23; 95%CI; 1.03–10.10; *p* = 0.044) compared to SUS group. Participants generally achieved satisfactory WL, regardless of healthcare systems. However, PVT participants had unfavorable lifestyle characteristics, highlighting the need for studies investigating environmental issues post-bariatric surgery.

## 1. Introduction

In 2019, more than 150 million Brazilians (71.5%) reported relying solely on the Unified Health System (SUS), the Brazilian universal public health system [1]. The SUS classifies metabolic bariatric surgery (MBS) as a complex treatment that requires specialized hospital units with pre- and postoperative follow-up routines. Conversely, the Brazilian private healthcare system (PVT) is mainly funded privately and follows different protocols. Despite being a well-founded model of public policy for a country with huge dimensions like Brazil, the SUS has structural problems that often affect service quality. The SUS comprises primary, secondary, and tertiary levels of care. The surgical procedure is at the tertiary level; however, patients are referred to the primary level of care for long-term clinical follow-up. After 2 or 3 years of surgery, patients often discontinue follow-up and remain without assistance from the SUS. In turn, the PVT system is funded directly by users or employers, mainly associated with private health insurance companies [2]. Thus, the economically most vulnerable population generally depends on the SUS, and people with better socioeconomic status use the PVT system, seeking more efficient assistance regarding structure, agility, and quality. Despite these structural differences, the health professionals of both services are specialized and well-trained. These two healthcare systems collectively perform around 118,000 MBS annually, placing Brazil second in the number of surgeries performed worldwide [3]. However, around 90% of these procedures are performed only by the private sector precisely because of the structural limitation of the SUS [2]. Considering that the population assisted by the SUS is economically more vulnerable, it is essential to comparatively analyze the therapeutic results between the Brazilian public and private healthcare services. The goal of obesity treatment is to maintain favorable long-term outcomes. Whilst consensus on a protocol to assess the response to MBS is lacking, studies commonly consider weight loss (WL) as the main result [4,5,6]. However, factors like improvement in health-related quality of life (HRQoL) [7,8] and lifestyle habits (physical activity level, diet quality, and alcohol consumption) [9,10,11] are also important and serve as markers of long-term results [12].

Both public and private healthcare services mandate pre- and postoperative follow-ups by a specialized multidisciplinary team following MBS. In the SUS, patients are followed by standardized specialized services until 18 months post-surgery when they return to primary care [13]. In contrast, PVT has no standard follow-up protocol, especially for the long term. Some studies have evaluated public and private Brazilian healthcare scenarios [14,15,16,17]. However, neither the validity and extent of implementation of the SUS care pathway nor its comparative therapeutic outcomes for MBS patients are known.

This study aimed to analyze the long-term response to Roux-en-Y gastric bypass (RYGB) for obesity beyond WL, assessing the lifestyle habits and HRQoL, comparatively, in patients using the public (SUS) and private (PVT) healthcare systems.

## 2. Materials and Methods

This cross-sectional study, performed as part of the “Food Consumption, Lifestyle, Control of Comorbidities and Nutritional Status of Patients Undergoing MBS” (CINTO) study, was conducted from July 2019 to March 2020. It included adults (18–65 years old) of both sexes living in the Federal District, Brazil, who underwent RYGB over at least five years. The SUS and PVT in the Federal District reflect the same national characteristics: the public sector has lower coverage than the private sector. While the SUS is centralized in a single hospital in the Federal District, the PVT performs MBS in 13 hospitals, but most of the population does not have the resources to access it.

Patients were recruited through social media advertisements and phone calls based on available SUS and PVT medical records. Those who underwent MBS or were followed up in public healthcare services were classified into the SUS group. Participants should have been operated on by surgeons as part of a multidisciplinary team. Pregnant or breastfeeding women and those unable to participate in the assessments were excluded. This study followed the Strengthening the Reporting of Observational Studies in Epidemiology (STROBE) reporting guideline for cross-sectional studies [18].

### 2.1. Demographics, Socioeconomic, Clinical, and Surgical Data

An electronic questionnaire collected information regarding demographics and socioeconomic data (age, sex, educational level, household and per capita income, and occupational status); surgical approach (presence of containment ring, postoperative time, and preoperative body mass index [BMI]); and clinical information (performance of plastic surgery, and pregnancy occurrence after MBS).

### 2.2. Study Variables and the Theoretical Model

We defined the healthcare system type (SUS/PVT) as the exposure variable and the response to surgical treatment as an outcome variable. Clinical and demographic characteristics were used as confounding variables, in addition to socioeconomic status (SES), a latent variable created using educational level, income per capita, and occupational status. Three markers were used to assess the response to surgical treatment: (1) WL and/or weight recurrence (WR), (2) lifestyle habits, and (3) HRQoL (Figure 1).

#### 2.2.1. Weight Loss and Weight Recurrence

Body weight and height were measured using standard procedures. Excess WL (%EWL) was calculated using a BMI of 25 kg/m^2^ [19]. Good responders were those who had the following WL performances: %EWL ≥ 50% and a BMI < 35 kg/m^2^ for individuals with preoperative BMI < 50 kg/m^2^, and %EWL ≥ 50% and a BMI < 40 kg/m^2^ for patients with preoperative BMI ≥ 50 kg/m^2^ [20]. WR was calculated from the nadir weight percentage post-surgery and considered relevant when greater than 10% [21].

#### 2.2.2. Lifestyle Habits

Lifestyle habits were assessed through diet quality, physical activity level, and alcohol consumption. Diet quality was defined by the processing degree of the food consumed. Three 24-h dietary recalls (24 hR) were applied on two non-consecutive weekdays and one day on the weekend, the first in person and the following telephonically. All recalls conducted were using the multiple-pass method [22]. The food preparations and beverages reported in the recalls were categorized into three groups from the NOVA classification: (*in natura*/unprocessed food, processed food, and ultra-processed food—UPF). NOVA is a food classification based on the nature, extent, and purpose of food processing, one of which group is specified as UPF food and drink products. The greater the degree of processing, the less healthy the diet, and, from this perspective, UPFs should not be consumed [23]. The amounts consumed were converted into grams, and the amount consumed in grams of food per 1000 kcal of energy consumed/day was calculated.

Similarly, physical activity level was assessed by 24-h physical activity recalls (24 hPAR) performed simultaneously with the 24 hR. On an hourly scale, the participants were to report the activities performed with their respective duration and intensity, starting, and ending at midnight. Metabolic equivalent of task (MET) values were assigned to each activity [24,25]. MET values averaged for each hour were added to obtain the total MET/day value. Subsequently, the measurement error of the total MET/day values obtained with the 24 hPAR was corrected using total MET/day values measured by a triaxial accelerometer (GT3X, Actigraph, Pensacola, FL, USA) in a subsample (n = 32) [26]. Physical activity was categorized into sedentary/insufficiently active (>1.0 and <1.6) and active/very active (>1.6 and <2.5) [27].

The Alcohol Use Disorders Identification Test (AUDIT) was used to screen for excessively harmful alcohol consumption [28]. Test responses were scored from 0 to 40. A total score ≥ 7 indicates a high risk of developing alcohol-related problems.

#### 2.2.3. Health-Related Quality of Life

The European Quality of Life 5 Dimensions 3 Level Version (EQ-5D-3L) was used to assess HRQoL [29]. It evaluates five dimensions (mobility, personal care, usual activities, pain/discomfort, and anxiety/depression) with three response levels (no problem, moderate problems, and extreme problems). Health status is defined by combining the level of each of the five dimensions, represented by a five-digit number. Thus, the EQ-5D-3L system defines 243 theoretically possible health states, and each health state generated can be converted into a unique EQ-5D-3L score or index that incorporates social preferences for health states. The EQ-5D-3L has been previously validated in the Brazilian population, which adopted a single index ranging from 1 to −0.17625 [30].

Additionally, respondents demarcated their health status on a visual analog scale (EQ-VAS) [29]. Respondents drew a line between the ‘box’ representing their state of health ranging from 0 to 100, considering 0 the worst and 100 the best imaginable health state.

### 2.3. Statistical Analysis

Descriptive analysis was performed by calculating absolute and relative frequencies for categorical variables and mean and standard deviation for quantitative variables. The Shapiro-Wilk Test was performed to verify the normality of quantitative variables. To consider the combined effect of socioeconomic variables, through the formation of the latent SES variable, a principal components analysis (PCA) was performed. A PCA allows for identifying larger and smaller sets of variables that stand out for use in a later multivariate analysis [31]. The SES was validated using exploratory analysis. The Kaiser-Meyer-Olkin index (KMO) was obtained to show the adequacy of the factor analysis, with values between 0.5 and 1.0 being considered acceptable. A factor loads greater than 0.3 and *p*-value < 0.05 indicated that the correlation between the observed variable and the constructor is moderately high. In the exploratory analysis, one main component was formed, contributing to 58.8% of the variance of the total information. The KMO index was 0.597, indicating a satisfactory value. The SES constituted the variables “educational level” (factor load = 0.843), “per capita income” (factor load = 0.800), and “occupation” (factor load = 0.639). This variable was categorized into three groups according to the tertile cut-off point.

The association between the study covariates according to the type of service (SUS/PVT) was evaluated using Student’s *t*-test to compare means, and Pearson’s chi-squared/Fisher’s exact test, to compare proportions. Covariates that showed statistically significant associations were utilized as adjustments in the analysis to assess the association between the type of service and outcome variables (response to surgical treatment).

To compare the quantitative variables according to the type of service, an analysis of covariance (ANCOVA) was used, adjusted by age, preoperative BMI, SES, and postoperative time. To compare the categorical variables (dependent variables) according to the type of service (SUS-reference category/PVT), the binary and multinomial logistic regression analyses were used, adjusted for age, preoperative BMI, SES, and postoperative time, except the variable “Individuals responders to WL,” which was not adjusted by preoperative BMI. The odds ratio with a 95% confidence interval was used to measure the effect.

The statistical program SPSS 23.0 (IBM Corp., Armonk, NY, USA) was used. Statistical significance was set at *p* < 0.05.

## 3. Results

A total of 516 individuals were screened for the study, and 123 participants were included in the analyses (Figure 2).

Table 1 shows demographics, socioeconomic, clinical, and surgical data. Individuals from the SUS group were comparatively older, with longer postoperative times and higher preoperative BMIs than the PVT group. The SUS group had a lower educational level, per capita income, and proportion of employed individuals, resulting in a higher proportion of individuals in the first tertile latent variable SES (G1) than the PVT group. Among the SUS group, there was a higher proportion of individuals with a lower socioeconomic level than those assisted by the PVT.

**Table 1 ijerph-20-06494-t001:** Demographics, socioeconomic, clinical, and surgical characteristics of the sample according to the healthcare system.

	TOTAL n = 123	SUS n = 79	PVT n = 44	*p*-Value
Age (years) (Mean ± SD)	48.3 ± 9.2	49.9 ± 9.0	45.6 ± 9.2	0.013 ^a^
Sex Female (n; %)	112 (91.1)	73 (92.4)	39 (88.6)	0.520 ^b^
Preoperative BMI (kg/m^2^) (Mean ± SD)	44.0 ± 8.2	46.0 ± 8.7	40.4 ± 5.8	<0.001 ^a^
Educational Level Years of study (Mean ± SD, min; max)	13.29 ± 4.0	12.00 ± 3.9	15.61 ± 2.9	<0.001 ^a^
Monthly per capita income (U.S. dollars—Mean ± SD)	480.1 ± 409.1	334.9 ± 296.9	740.8 ± 454.5	0.002 ^a^
Employed (n; %)	84 (68.2)	49 (62.0)	35 (79.5)	0.045 ^c^
SES (latent variable)				
Tertile 1 Tertile 2 Tertile 3	41(33.3) 41(33.3) 41(33.3)	36 (45.5) 30 (38.0) 13 (16.5)	5 (11.4) 11 (25.0) 28 (63.6)	<0.001 ^c^
Postoperative time (years) (Mean ± SD)	9.3 ± 2.6	9.8 ± 2.6	8.50 ± 2.2	0.005 ^a^
Surgical Approach (n = 122)				
Open Laparoscopic	83 (67.5) 39 (31.7)	57 (73.1) 21 (26.9)	26 (59.1) 18 (40.9)	0.112 ^c^
Presence of contention ring (n; %)	53 (43.1)	37 (46.8)	16 (36.4)	0.261 ^c^
Reported pregnancy(ies) after MBS (n = 112) (n; %)	12 (9.8)	5 (6.8)	7 (18.4)	0.102 ^b^
Underwent plastic surgery after MBS (n;%)	70 (56.9)	46 (58.2)	24 (54.5)	0.693 ^c^

^a^ Student’s *t*-test; ^b^ Fisher’s exact test, ^c^ Pearson’s chi-squared test. BMI = body mass index, MBS = Metabolic and bariatric surgery, PVT = individuals who underwent bariatric surgery and were followed up in the private healthcare service, SES = socioeconomic status (composed of the variables educational level, per capita income, and occupation), SUS = individuals who underwent bariatric surgery or who were followed up in the public healthcare service. Regarding the therapeutic response to surgery, the %EWL, processed food intake, and HRQoL were higher in the PVT group; however, these differences were not statistically significant after adjustments (Table 2).

**Table 2 ijerph-20-06494-t002:** Comparison of means of quantitative markers of response to surgical treatment of obesity (weight, lifestyle habits, and health-related quality of life) according to healthcare system type.

	TOTAL n = 123	SUS n = 79	PVT n = 44	*p*-Value	
				Unadjusted	Adjusted
Weight markers (Mean ± SD)					
% Total Weight Loss ^1^	24.64 ± 0.99	24.71 ± 1.32	24.5 ± 1.41	0.929	0.459
% Excess Weight Loss ^2^	60.46 ± 2.41	56.88 ± 3.03	66.89 ± 3.83	0.046	0.219
% Weight Recurrence ^3^	20.42 ± 1.46	20.89 ± 2.09	19.60 ± 1.64	0.675	0.384
Diet quality ^4^ (Mean ± SD)					
Consumption (grams/1000 kcal) of:					
*In natura*/Unprocessed foods Processed foods Ultra-processed foods	637.53 ± 181.37 113.68 ± 50.74 195.65 ± 110.00	660.1 ± 193.21 103.43 ± 41.72 186.94 ± 106.57	597.09 ± 151.70 132.10 ± 60.15 211.30 ± 115.49	0.065 0.002 0.241	0.395 0.093 0.645
Health-related quality of life (n = 111) (Mean ± SD)					
Self-perception of current health (EQ-VAS) ^5^ EQ-5D-3L index ^6^	70.09 ± 21.54 0.70 ± 0.17	66.90 ± 23.37 0.67 ± 0.19	75.75 ± 16.66 0.76 ± 0.14	0.013 0.037	0.797 0.375

EQ-5D-3L = European Quality of Life 5 Dimensions 3 Level instrument; EQ-VAS = Visual Analogue Scale of European Quality of Life instrument; PVT = individuals who underwent bariatric surgery and were followed up in the private healthcare service; SD = standard deviation; SUS = individuals who underwent bariatric surgery or who were followed up in the public healthcare service. a = Student’s *t*-test; b = analysis of covariance (ANCOVA) adjusted for age, preoperative body mass index, socioeconomic status, and postoperative time. ^1^ Percentage of Total Weight Loss = [current weight (kg) − preoperative weight (kg)]/current weight (kg) × 100; ^2^ Percentage of Excess Weight Loss = [current weight (kg) − preoperative weight (kg)]/[preoperative weight (kg)] − weight considering Body Mass Index of 25 kg/m^2^ (kg)] × 100; ^3^ Percentage of Weight Recurrence = percentage from the nadir weight after surgery greater than 10% [22]; ^4^ Quality of diet = evaluated by three 24-h food recalls (24 hR), and foods were categorized into three groups, according to the criteria of the classification proposed by Monteiro et al. (2018) [32]; ^5^ Self-perception of current health through an EQ-VAS scale from 0 to 100 (0, the worst and 100, the best imaginable health state); ^6^ Score obtained from the EQ-5D-3L based on social preferences for health states and validated for the Brazilian population, ranging from −0.17625 to 1, where 1 represents the best state of health.

Concerning the HRQoL domains, Figure 3 shows that, in the pain/discomfort and anxiety/depression domains, more than half of the sample reported at least moderate problems, however, without differences between groups by adjusted multinomial regression.

Table 3 shows the data of only the participants who achieved the expected therapeutic responses during data collection. Participants in the PVT group were better responders to excess WL only before adjustment for potential confounders. Nearly one-third of participants had a high risk of developing alcohol-related problems, which was significantly higher in the PVT group compared to the SUS group (45.9% vs. 24.6%). Controversially, the PVT group had a higher chance of being physically inactive than the SUS group. 

## 4. Discussion

This study comprehensively evaluated patients at a late stage in the surgical treatment of obesity from both SUS and PVT healthcare systems. In general, satisfactory results were observed for most markers and similarly for both healthcare modalities. As expected, patients who underwent MBS in the SUS group exhibited lower SES than those who underwent surgery in the private system. Also, the SUS group presented a higher preoperative BMI when compared to the PVT group, and this is possibly due to the public system cannot quickly attend to all patients who need bariatric surgery. Thus, the SUS prioritizes the most critically ill patients with the highest BMI. In addition, for the same reason, patients wait a long time before being able to undergo surgery in SUS, leading to weight gain during this waiting period. However, there was likely no difference in weight loss between groups because we are considering a percentage of weight loss in the long-term post-surgery, after a period of weight stabilization. On the other hand, the PVT group individuals were more prone to develop alcoholism-related problems and were less likely to have an adequate physical activity level. 

The long-term results of surgical obesity treatments must also be evaluated from the perspective of the patients’ SES, which different variables can address. Our team recently published a systematic review with a meta-analysis that analyzed the association between SES and late postoperative WL [33]. The meta-analysis revealed a potential influence of race/ethnicity on socioeconomic factors, where black individuals were less likely to lose weight. This likely contributes to the definition of SES in the Brazilian population since most users of SUS are non-white [34]. Considering the multivariate long-term benefits of MBS besides WL *per se* [12,35], this study addresses confounding factors to avoid errors in interpreting all results and comparisons, especially when SES aspects are involved.

Regarding physical activity levels, some studies show that individuals with lower SES perceive more barriers to changing lifestyle habits, probably given their lower knowledge regarding what a healthy lifestyle entails [36,37]. However, in the present study, SUS participants were more physically active, possibly due to urban mobility issues and the need to walk longer distances to work and other daily activities compared to private service users, as also observed in developed countries [38,39]. 

Regarding alcohol consumption, the increased risk of developing problems with alcohol among patients undergoing MBS is a complex issue involving psychological and physiological mechanisms that are not yet fully understood. Furthermore, some studies in rats already show that RYGB promotes new-onset alcohol intake, which can be explained by the reduction in gastric ghrelin secretion and subsequent malfunctions of ghrelin-1a receptor (GHSR) signaling. The hypothesis is that the surgery causes primarily decreased appetite and indirectly stimulates new onset alcohol intake [40,41,42,43]. Even so, the expressive prevalence of high risk for the development of alcohol problems in the PVT group, compared to the SUS group found in our sample, requires better elucidation.

HRQoL must be considered a potential outcome since health encompasses, among others, the physical, social, and mental aspects [44]. MBS offers a persistent benefit in terms of overall health condition and HRQoL, especially for its physical component score [8], as reported in studies conducted in developed [12,45] and developing countries [17,46]. The EQ-5D-3L has the advantage of analyzing the HRQoL of individuals considering each associated domain and its self-perception. This study also revealed a high prevalence of patients with moderate problems of anxiety/depression and pain/discomfort when analyzing the domains separately. These constitute loci of attention that may impair HRQoL, regardless of SES or the healthcare system. Recently, researchers investigated whether postoperative patients ‘quality of life could be predicted using baseline data by applying the QALIS (quality-adjusted life years) instrument [47]. Characteristics including health-related quality of life, age, sex, BMI, postoperative complications within six weeks, and smoking status may be adequate in predicting their postoperative QALYs after one year. However, this association between preoperative and long-term postoperative periods has not yet been evaluated.

Another pillar element of the lifestyle concerns diet quality. Although no difference in the dietary pattern was observed between groups, habitual consumption of processed food and UPFs was apparent. Diets high in UPFs tend to contain greater intakes of energy, free sugars, total and saturated fats, and lower intakes of fiber, protein, and some micronutrients [48]. Furthermore, UPF consumption may be associated with a higher risk for postprandial hypoglycemia, which may also interfere with weight loss results [49]. Thus, the dietary pattern should also be improved, despite the positive markers found, to avoid adverse health-related outcomes associated with UPF consumption [50]. In the first months after surgery, there is a tendency to make better food choices with reduced consumption of processed food and UPFs [11,51]. However, some studies showed a tendency to return to the same eating habits as before the surgical procedure [52,53]. Farias et al. (2020) conducted a prospective study with a 24-month follow-up of 32 patients after RYGB in a public healthcare system in Paraná, Brazil. They found an increase of 60.0% in the calorie consumption of processed food and UPFs between 6 and 24 months postoperatively, and the consumption of processed food and UPFs exceeded 50% of the total calorie intake of the diet in all periods analyzed [11]. These findings reinforce the need to maintain follow-up of these patients with nutritional counseling, aiming not only at controlling energy intake but mainly at its qualitative aspects.

This study had some limitations. The cross-sectional design does not allow the establishment of a cause-and-effect relationship. The recruitment criteria may introduce selection bias since we did not reach a representative sample. However, the sample size reached power above 80%. Additionally, the small percentage of males in the sample, the absence of some information related to the environment (family composition, religiosity, availability of local food stores, and presence of safe places to practice physical activities), as well as other potential confounding factors not addressed in the analyzes related to the health systems characteristics, limited the inference power of the analyzes. However, this study yields therapeutic results later in treatment using specialized markers beyond body weight in a group with different SESs than analyzed in other studies. Statistical scrutiny using a theoretical model of analysis and control of confounding variables further reinforces the validity of this study.

## 5. Conclusions

Although the WL outcomes were, on average, satisfactory for the total sample, individuals assisted in the PVT sector had a less healthy lifestyle, a greater risk of developing alcohol problems, and a lower level of physical activity. Follow-up of these patients is recommended with special attention to the risk of poor diet quality in the long term after MBS. Investigating the determinants of these differences between SUS and PVT healthcare systems is still necessary. The results suggest that the SUS can play its role in treating severe obesity and reinforce the need to maintain investments in national public policies.

## Figures and Tables

**Figure 1 ijerph-20-06494-f001:**
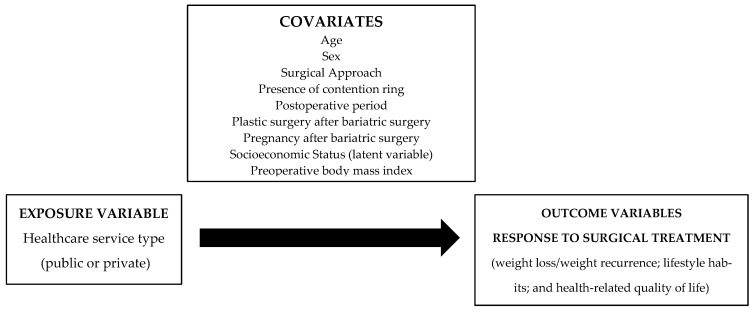
Theoretical model for defining the research gap and variables for choosing the statistical analysis method. Lifestyle habits included physical activity level, diet quality, and alcohol consumption. Socioeconomic status is a latent variable created using educational level, income per capita, and occupational status.

**Figure 2 ijerph-20-06494-f002:**
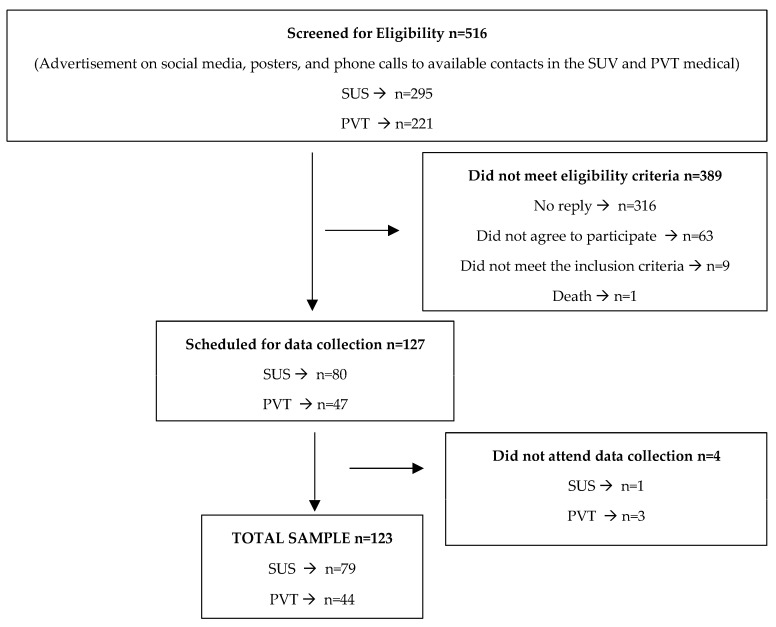
Flowchart of participants in the CINTO study and final sample. PVT = individuals who underwent bariatric surgery and were followed up in the private healthcare service, SUS = individuals who underwent bariatric surgery or who were followed up in the public healthcare service.

**Figure 3 ijerph-20-06494-f003:**
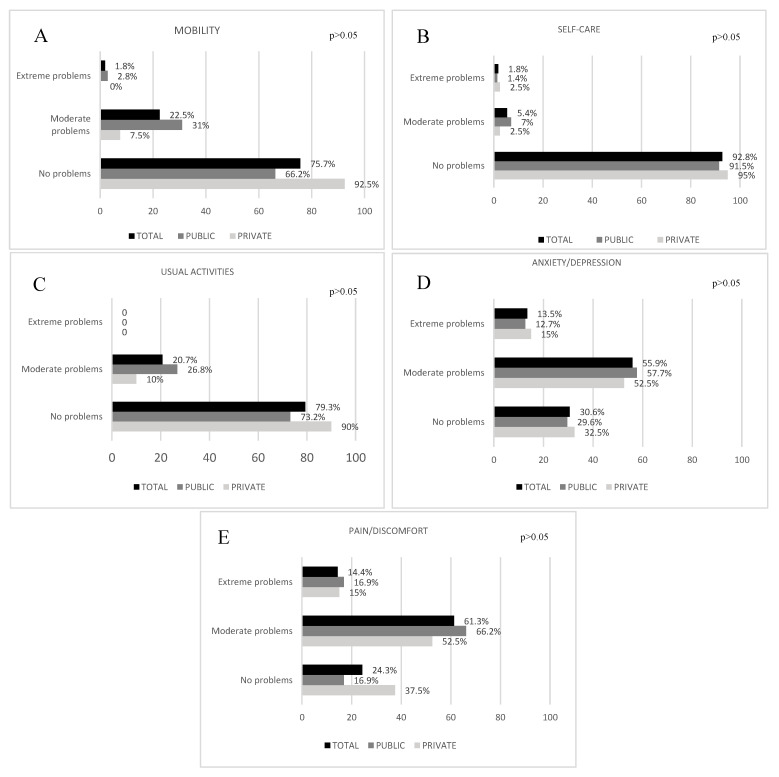
Percentage of individuals (total sample, public and private group) with no problems, moderate or extreme problems by the five EQ-5D-3L domains: (**A**) = Mobility, (**B**) = Self-care, (**C**) = Usual activities, (**D**) = Anxiety/Depression, (**E**) = Pain/Discomfort. *p* > 0.05 for all comparisons in all domains according to Multinominal Regression adjusted by age, postoperative time, preoperative body mass index, and socioeconomic level (n = 111).

**Table 3 ijerph-20-06494-t003:** Percentage distribution of the sample according to healthcare system type and models (simple and multiple) of binary logistic regression to evaluate the qualitative markers of response to obesity surgical treatment.

	TOTAL n (%)	SUS n (%)	PVT n (%)	OR (95% CI)		*p*-Value	
				Unadjusted	Adjusted	Unadjusted	Adjusted
Weight markers (n = 123)							
Individuals with weight recurrence ^1^	92 (74.8)	57 (72.2)	35 (79.5)	1.50 (0.62; 3.63)	0.90 (0.30; 2.70)	0.367	0.850 ^a^
Individual responders to weight loss ^2^	81 (65.9)	47 (59.5)	34 (77.3)	2.32 (1.00; 5.34)	1.25 (0.47; 3.36)	0.049	0.655 ^b^
Lifestyle habits							
Alcohol Consumption ^3^ (n = 106)							
High risk of developing alcohol-related problems (AUDIT)	34 (32.08)	17 (24.6)	17 (45.9)	2.60 (1.11; 6.07)	3.23 (1.03; 10.10)	0.025	0.044 ^a^
Physical Activity Level ^4^ (n = 122)							
Sedentary/Insufficiently Active Very active	41 (33.6) 81 (66.4)	19 (24.4) 59 (75.6)	22 (50.0) 22 (50.0)	1 0.32 (0.15; 0.71)	1 0.23 (0.87; 0.63)	0.004	0.004 ^a^

CI = confidence interval; OR = odds ratio; PVT = individuals who underwent bariatric surgery and were followed up in the private healthcare service; SUS = individuals who underwent bariatric surgery or who were followed up in the public healthcare service. ^a^ Model of binary logistic regression adjusted for age, preoperative BMI, socioeconomic status (SES) and postoperative time. ^b^ Adjusted for age, socioeconomic status (SES) and postoperative time; the independent variable of all models was healthcare service type (public as categorical reference). ^1^ Individuals with percentage from the nadir weight after surgery less than 10% [21]; ^2^ Responder to weight loss = individuals with preoperative BMI < 50 kg/m^2^ → %EWL ≥ 50% and a BMI < 35 kg/m^2^, individuals with preoperative BMI ≥ 50 kg/m^2^ → %EWL ≥ 50% and a BMI < 40 kg/m^2^ [20]; ^3^ Alcohol Consumption = evaluated by applying the Alcohol Use Disorders Identification Test (AUDIT) [28]. Total score ≥ 7 → high risk of developing alcohol-related problems; ^4^ Physical Activity Level = assessed by applying three 24-h physical activity recall, classified, and expressed as metabolic equivalent of task MET per hour/day according to Saraiva Leão et al. protocol [26]. Then, physical activity was categorized as sedentary/insufficiently active (>1.0 and <1.6) or active/very active (>1.6 and <2.5) [27].

## Data Availability

The data presented in this study are available on request from the corresponding author.

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
