# Peer review of "Long-Term Lifestyle Habits and Quality of Life after Roux-in-Y Gastric Bypass in Brazilian Public versus Private Healthcare Systems: Beyond Weight Loss"

_ijerph, 2023, doi:10.3390/ijerph20156494_

Round 1
Reviewer 1 Report
The manuscript is focused on a difference in outcome of bariatric surgery based on the choice of public versus private hospital.
The Abstract is concise and well structured.
The Introduction is giving sufficient background information on the topic.
The Materials and Methods are comprehensively written. With the study using questionnaires, my only concern is the accuracy of self reported data regarding objective variables such as their current weight. Additionally, the study would be interesting as a design for a larger sample size analysis.
Results section with a very structured presentation:
- Table 1 : use the width of the entire page
Discussion - well anchoring the results in the scientific literature available on the topic.
Conclusion - supported by the study outcomes.
In my opinion, the manuscript meets the standards of publication and should be accepted in present form.
Regards,
Your Reviewer
Author Response
- With the study using questionnaires, my only concern is the accuracy of self-reported data regarding objective variables such as their current weight.
Thanks for your thoughtful review and for all your contributions. Regarding to data collection, in general, we did not used self-reported data, but measured data. The only self-reported data was obtained from preoperative weight, that we believe to be reliable, since this data is highly reinforced and present in the memory of people undergoing bariatric surgery.
- Additionally, the study would be interesting as a design for a larger sample size analysis.
Our comments: In fact, it would have been valuable if we had achieved a larger sample, but it is always difficult to attract volunteers. Additionally, we had to interrupt data collection due to the start of the pandemic in March 2020. Even so, the sample power was sufficient to present these analyses, albeit with some limitations common in cross-sectional studies.
Results - Table 1 : use the width of the entire page
Our comments: We adjusted the formatting as suggested. Thank you.

Reviewer 2 Report
This is a well-written study on the outcomes following RYGB in patients treated in SUS and PVT healthcare systems.
The statistical analyses used are solid, the results presented are clear, and adeguately discussed by the authors, who also have adeguately explained the limitations of their study. I only have some minor suggestions.
Line 109: Has WR been already defined? if not please open.
Table 1, please round the age entry.
Had BMI a normal distribution? (if not, as typically is the case, please show as median [IQR])
What type of intervention the patients received? If they only received RYGB, perhaps this should be made clear already in the title (substitute bariatric surgery with RYGB).
Line 324: to what they is referred to. Perhaps easier for the reader to clarify.
Lines 340-341: UPFs probably will be related with higher risk for post-prandial hypoglycaemia (PPHG). Is there any evidence on the matter? At least one previous study on BS patients has suggested that PPHG may lead to weight regain (Obes Surg. 2020 Jun;30(6):2266-2273. doi: 10.1007/s11695-020-04465-9) which might have also contributed to your findings of no difference between the 2 groups.
Another interesting aspect is that the preoperative BMI in the SUS group was higher, and it would have been expected that these people would have lost more weight at Follow-up (proportionally to the higher BW at baseline). If the authors wish, they could elaborate a bit more their thoughts why this was not the case here.
Author Response
REVIEWER 2:
- Line 109: Has WR been already defined? if not please open.
Our comments: In fact, there are many available criteria for defining weight recurrence (WR) cited in line 82/ Methods. Until now, there is no standard definition for clinically significant weight recurrence after bariatric surgery1. However, in our study, WR was calculated from the nadir weight percentage post-surgery and considered relevant when greater than 10%, since it is an usual cut-off point and has already been used as a reference in previous publications 1,2.
1 Majid, S.F.; Davis M.J.; Ajmal, S.; Podkameni, D.; Jain-Spangler, K.; Guerron, A.D.; King, N.; Voellinger, D.C.; Northup, C.J.; Kennedy, C.; Archer, S.B. Current state of the definition and terminology related to weight recurrence after metabolic surgery: review by the POWER Task Force of the American Society for Metabolic and Bariatric Surgery. Surg Obes Relat Dis. 2022 Jul;18(7):957-963. doi: 10.1016/j.soard.2022.04.012. Epub 2022 Apr 29.
2 Da Silva, F.B.; Gomes, D.L.; De Carvalho, K.M. Poor diet quality and postoperative time are independent risk factors for weight regain after Roux-en-Y gastric bypass. Nutrition. 2016 Nov-Dec;32(11-12):1250-3. doi: 10.1016/j.nut.2016.01.018) .
- Table 1, please round the age entry.
Our comments: We correct accordingly.
- Had BMI a normal distribution? (if not, as typically is the case, please show as median [IQR])
Our comments: Yes, BMI had a normal distribution by the Shapiro Wilk test (performed to verify the normality of the quantitative variables). We added in methods section that we performed this normality test to make it clearer (lines 154-156).
- What type of intervention the patients received? If they only received RYGB, perhaps this should be made clear already in the title (substitute bariatric surgery with RYGB).
Our comments: All patients underwent RYGB, therefore, we agreed with the reviewer and the title of the manuscript became “Long-term lifestyle habits and quality of life after Roux-in-Y Gastric Bypass in Brazilian public versus private healthcare systems: beyond weight loss”.
- Line 324: to what they is referred to. Perhaps easier for the reader to clarify.
Our comments: Thanks for the observation. In fact the sentence was not clear. We remove the pronoun "they" from the sentence. (line 317)
- Lines 340-341: UPFs probably will be related with higher risk for post-prandial hypoglycaemia (PPHG). Is there any evidence on the matter? At least one previous study on BS patients has suggested that PPHG may lead to weight regain (Obes Surg. 2020 Jun;30(6):2266-2273. doi: 10.1007/s11695-020-04465-9) which might have also contributed to your findings of no difference between the 2 groups.
Our comments: Thank you very much for your suggestion. In fact, the consumption of UPF may be related to more episodes of reactive hypoglycemia interfering with the weight loss performance, however, it would not be possible to assess this issue in our analysis, since we did not include those variables in our protocol. In any case, we added this hypothesis in the discussion section, due to its relevance for future studies to strengthen the scientific evidence on this matter. (lines 334-335)
- Another interesting aspect is that the preoperative BMI in the SUS group was higher, and it would have been expected that these people would have lost more weight at Follow-up (proportionally to the higher BW at baseline). If the authors wish, they could elaborate a bit more their thoughts why this was not the case here.
Our comments: The population assisted in the SUS group had a higher BMI than patients from the PVT since the public system cannot quickly attend to all patients who need surgery. Thus, the SUS prioritizes the most critically ill patients, with the highest BMI. In addition, for the same reason, patients wait a long time before being able to undergo surgery in SUS and are likely to gain weight during the waiting period. However, it is likely they did not lose proportionally more weight than those patients in the PVT because we are considering a percentage weight loss in the long-term post-surgery, after a period of weight stabilization. We have inserted a paragraph in the discussion to comment on this aspect (lines 293-298).

Reviewer 3 Report
Congratulations to the authors for presenting their long-term results on the quality of life after bariatric surgery. Both long-term studies and assessment of the quality of life itself are interesting and important topics in the bariatric literature and are generally lacking. The authors have further incorporated subjects such as socioeconomic status, quality of food, and alcohol consumption, which add additional leverage to their research.
The limitations of the study, mostly owing to study design, have been adequately addressed.
In the attached file you can find my specific comments, which mostly revolve around the accepted standardized terminology in contemporary bariatric literature.

Author Response
REVIEWER 3:
1. Line 19: at least 2 metrics should be mentioned, whereas a complete report (including itial mean cohorft BMI, _BMI) is highly recommended. Our comments: We agree on the importance of this information in abstract. However, due to limited length of the abstract it was not possible to include all of these weight loss markers. As pointed out by the reviewer, at least %EWL was included in the abstract section (line 19). Regarding preoperative BMI, this information is only shown in Table 1, in order not to exceed the number of words allowed in the abstract. 2. Line23: If this is the case, then where does the p=0.002 value that you mention immediately ealier refer to?Our comments: p value referred to the difference between processed food ingestion between the PVT and the SUS group before adjustments, in fact, we removed the p-value cited by the reviewer because it was an error. We apologize for that.3. Line 24: Standard referecing is as follows: 95% CI xx.x
Our comments: thank you. we correct accordingly.
4. Line 33: The comtemporary term is “metabolic bariatric surgery”, stressing out that the objective is not solely weight loss, but restoration of the health status as a whole, in the context that obesity is treated as a multifactorial and complex disease rather than a temporary condition.Our comments: We totally agree and have changed the term “Bariatric Surgery” to “Metabolic Bariatric Surgery” throughout the manuscript, as suggested by the reviewer. 5. Line 63: Doesn’t this mode of recruitment introduce selection bias?Our comments: It is possible that this recruitment procedure caused bias, since we do not achieve a representative sample. However, considering the difficulties in recruiting people to participate in these types of studies, we decided to use different methods for select volunteers, despite the limitations and biases generated by this procedure. We have added this topic to the discussion section of the manuscript (line 346). 6. Table 1: How is it possible that total and SUS income are exactly the same? The other group population is not negligent.Our comments: We would like to clarify that Total refers to the total sample, and SUS monthly per capita income were not the same. We believe that there must have been some modification in the formatting of the pdf document that changed the correct visualization of the data. TOTAL = 480.1±409.1; SUS=334.9±296.9; and PVT=740.8±454.5. 7. Table 1: Since you have such a long term follow-up period, why didn’t you chose to use the QALYs as a surrogate measure of the quality of life?
Our comments: Indeed, the QALYs could be another possibility of instrument for assessing the health-related quality of life. Despite actually having relevant properties, it is based on UK population and so far there was no validated version for the Portuguese language. Considering that our study is cross-sectional, and we do not have enough baseline data to allow us to apply the QALYs, the EQ-5D-3L was properly applied for our study. Despite some limitations inherent to the instrument, the EQ-5D-3L also addresses health-related quality of life and has been validated for our language. In any case, we mentioned this instrument in the discussion of the article, to draw the readers' attention to other instruments that need to be applied in future studies to deepen the analysis of this topic (lines 326-329).
8. Line 329: A recent relevant study, based on the Swedish nationwide quality register revealed the predictive value of preoperative characteristics with the QALYs by means of logistic regression models. You might consider citing it. Our comments: This recently published study (Sun et al, 2023) is interesting and opens up the possibility of predicting health-related quality of life after bariatric surgery based on baseline characteristics, especially in the first two years of surgery. We have inserted a comment about this in our discussion (lines 326-329).
